# The Determinants of Vaccine Literacy in the Italian Population: Results from the Health Literacy Survey 2019

**DOI:** 10.3390/ijerph19084429

**Published:** 2022-04-07

**Authors:** Chiara Cadeddu, Luca Regazzi, Guglielmo Bonaccorsi, Aldo Rosano, Brigid Unim, Robert Griebler, Thomas Link, Paola De Castro, Roberto D’Elia, Valeria Mastrilli, Luigi Palmieri

**Affiliations:** 1Department of Life Sciences and Public Health, Università Cattolica del Sacro Cuore, 00168 Rome, Italy; chiara.cadeddu@unicatt.it (C.C.); luca.regazzi01@icatt.it (L.R.); 2Department of Health Sciences, University of Florence, 50134 Florence, Italy; guglielmo.bonaccorsi@unifi.it; 3National Institute for the Analysis of Public Policy, 00198 Rome, Italy; a.rosano@inapp.org; 4Department of Cardiovascular, Endocrine-Metabolic Diseases and Aging, Italian National Institute of Health, 00162 Rome, Italy; brigid.unim@iss.it; 5Competence Centre Health Promotion and Health System, Austrian National Public Health Institute, A-1010 Vienna, Austria; robert.griebler@goeg.at (R.G.); thomas.link@goeg.at (T.L.); 6Scientific Communication Unit, Italian National Institute of Health, 00161 Rome, Italy; paola.decastro@iss.it; 7Ministry of Health, Viale Giorgio Ribotta, 5, 00144 Rome, Italy; r.delia@sanita.it (R.D.); valeria.mastrilli@aslroma2.it (V.M.)

**Keywords:** health literacy, vaccination, health knowledge, attitudes, practice, health promotion, vaccination hesitancy, information-seeking behavior, surveys and questionnaires, COVID-19

## Abstract

Vaccines are among the most important public health achievements of the last century; however, vaccine awareness and uptake still face significant challenges and the COVID-19 pandemic has only exacerbated this phenomenon. Vaccine Literacy (VL) is the ability to find, understand and judge immunisation-related information to make appropriate immunisation decisions. A cross-sectional study on a sample of 3500 participants, representative of the Italian adult population aged 18+ years, was conducted in Italy in 2021. A validated questionnaire, including sections on health literacy (HL), sociodemographic characteristics, risk factors, and lifestyles of respondents, was used. VL was measured by four items (item 19, 22, 26 and 29) of the HL section. While 67.6% of the respondents had a “good” (47.5%) or “sufficient” (20.1%) level of VL, 32.4% had “limited” VL levels. Although the overall VL level was quite high, many participants reported difficulties in dealing with vaccination information, particularly those with a lower educational level, those living in southern and insular regions of Italy, those with greater financial deprivation and those with a migration background. Improving VL in Italy should be a top priority in the political agenda, with special regard to socially and geographically disadvantaged communities.

## 1. Introduction

Vaccines are among the most important medical achievements over the past century, preventing millions of illnesses, disabilities, and deaths worldwide, and are considered one of the most cost-effective public health interventions [1]. However, vaccine awareness and uptake face significant challenges, and the COVID-19 pandemic has put additional pressures on governments to deploy vaccination programmes and maintain public trust in vaccines and the scientific community, which has proven crucial in Italy in recent years [2,3]. Variations in vaccination coverage across and within countries reveal inequalities in vaccine uptake and information that can lead to unvaccinated populations causing outbreaks of vaccine-preventable diseases. Understanding and addressing such inequalities is key to increase overall vaccination coverage and ensure greater equity in health outcomes at regional, national and European levels.

Italy, in particular, faces the threat of hesitancy due to sociocultural heterogeneity, which led the country into a severe measles outbreak in 2017 [4] and contributed to a total of five million Italians who are still unvaccinated against COVID-19 more than 1 year after the start of the mass vaccination campaign [5]. 

Recently, Geiger et al. [6] reported that factors affecting vaccination coverage are mainly represented by vaccination availability and vaccination readiness, which were measured on a seven components scale (confidence, complacency, constraints, calculation, collective responsibility, compliance and conspiracy). 

This model highlights the importance of identifying and understanding the sociocultural factors behind the reasons to vaccinate or not vaccinate. Research on determinants of vaccination/non-vaccination has shown that decisions for or against immunisation (i.e., vaccination behavior) are driven by individual and collective experiences and beliefs, knowledge and situational/contextual conditions that depend not only on issues related to the individual, but also on healthcare professionals and their attitudes, as well as on the specific vaccine being considered [6,7,8,9,10].

A recent systematic review identified nine studies (mostly conducted in the United States and in countries other than Italy) that examined the relationship between Health Literacy (HL) and vaccination behavior and the attitudes toward vaccination. Despite partially contradictory results, the studies indicate a positive correlation between HL and vaccination attitudes and behavior, with higher HL associated with a more positive attitude toward vaccination and greater uptake of vaccinations [11]. An updated review of the literature, which was included in the European Health Literacy Population Survey 2019–2021 (HLS_19_) International Report, indicated similar partially conflicting results and hypothesised a positive correlation between HL and vaccinations perceived as relevant (e.g., influenza for older people, human papillomavirus for younger women), suggesting that individuals with higher HL could identify vaccinations that were more relevant to them while skipping others [12].

While HL, in general, remains a topic of great interest [13,14,15], the growing evidence in favor of a strong relationship between HL and vaccination behavior has led to the development of a relatively new research topic known as vaccine literacy (VL) [16,17,18,19,20,21]. In this article, following the HLS-EU Consortium’s definition of general HL and in analogy with other definitions of vaccine literacy, the adopted definition of VL consists in “people’s knowledge, motivation and skills to find, understand and evaluate immunisation-related information in order to make adequate immunisation decisions” [12,16,17,22]. However, next to the definition used in this study, other statements which present many similarities, but also some—although small—differences or partial different points of view, can be found. 

The definition of VL used in this article was raised from the work of Sorensen et al. on HL [22], which already foresaw a deep review of the definitions appeared since 1998, with the publication of Nutbeam’s definition in the Health Promotion Glossary [23], and was applied on the vaccine/vaccination dimension. The VL definition adopted in this study, evokes the four key sub-dimensions that influence vaccination behavior and that appear to be even more significant in the COVID-19 era to contain the spread of the pandemic: find/obtain/access information on recommended vaccinations for individuals and their family; understand why individuals or their family may need vaccinations; judge/appraise/evaluate the need of vaccinations for individuals and their family; and apply/use the information to decide whether individuals should be vaccinated.

Other examples of VL definitions can be found in the recent literature: in the definition adopted by Michel and Goldberg [18], the accent is posed on exploring the vaccine-decision process to identify interventions that can positively influence vaccine uptake; further back in time, the definition adopted by Ratzan [17] considered, other aspects of VL regard advocacy and the development of a system with decreased complexity to communicate and offer vaccines as *conditio sine qua non* of a functioning health system, aimed at creating a modification of in the social norm for advancing vaccine uptake, and providing herd immunity with a foundation of vaccine/health literacy at a level commensurate with age, mental capacity, gender and environment.

There has been scarce research to date on the VL of the Italian population. Only three studies have been performed with this aim, two of which were related to specific vaccines and/or specific target groups (COVID-19 and influenza in nursing home staff) [19,20] and one was a pilot study to assess a new measurement tool, that provided incomparable results and limited samples [16]. More studies have been undertaken worldwide, especially regarding the relationship between vaccine literacy, vaccine hesitancy and the acceptance of COVID-19 vaccines [24,25,26,27,28,29,30].

Since 2018, Italy has participated in the Action Network on Measuring Population and Organizational Health Literacy (the M-POHL network) under the umbrella of the World Health Organization European Health Information Initiative (WHO-EHII), in alignment with Health 2020, the European policy framework for health and well-being. In the framework of participation in the M-POHL network, the HLS_19_ [12] was implemented in 17 countries of the WHO European Region. In Italy, the National Institute of Health (ISS), with the support of the Ministry of Health, coordinated the country’s participation in the M-POHL network and the implementation of the HLS_19_ in the Italian population.

The present study investigated the VL of a large representative sample of the Italian population and its relationship with HL, as well as with demographic, socioeconomic and health-related variables.

## 2. Materials and Methods

A cross-sectional study of 3500 participants aged 18 years and over, representative of the Italian population, was conducted using an online and telephone survey from 8 April to 8 May 2021. The HLS_19_ was implemented in Italy as part of an international collaboration initiated by the WHO-EHII affiliated M-POHL network [12]. The HLS_19_-Q47, an updated version of the European Health Literacy Questionnaire (HLS-EU-Q47) developed in the framework of the M-POHL collaboration was administered to the Italian population and included a 47-item section specifically focusing on health literacy (HL), a 34-item section on the sociodemographic characteristics of the respondents, and a 16-item section on coronavirus-related health literacy issues. Four items (items 19, 22, 26 and 29, as shown in Table 1) of the HLS_19_-Q47 measured the VL scale, investigating the four aspects of vaccination-related information management (see “Sub-dimensions” in Table 1): to find/access/obtain, understand, judge/appraise/evaluate, and decide/apply/use information relevant for vaccination behavior. The VL scale was validated internationally: Cronbach’s alpha showed its high reliability; confirmatory factor and discriminant analyses revealed that the VL scale measures a different but related trait than the HL scale; and, finally, the overall data-model fit to the Rasch model was deemed sufficient [12].

For all VL items, a 4-point Likert scale was applied with the following categories: “very easy”, “easy”, “difficult”, and “very difficult”. In some countries but not Italy, an optional package of nine items specific to health literacy and vaccination was also administered [12].

The VL score was calculated as the percentage (range 0–100) of items with valid responses that were answered “very easy” or “easy” and was determined only for respondents with complete data for the four VL items. A higher score represented a higher VL level [12].

The difficulty of each VL item was evaluated as the percentage of the “very difficult” or “difficult” responses combined. For each item, the proportions of respondents ticking each of the four response categories are reported in Table 1.

In analogy with the categorisation adopted by Röthlin and colleagues in a similar context [31], individual VL levels were assigned to the respondents according to the following definitions for the cutoffs:
■Good: “very easy” + “easy” = 100.0% (4 out of 4 answers); ■Sufficient: “very easy” + “easy” = 75.0% (3 out of 4 answers); ■Limited: “very easy” + “easy” < 75% (fewer than 3 out of 4 answers).

The HL score was based on a subset of 12 items (HLS_19_-Q12) of the HLS_19_-Q47, which did not include the four vaccination items. Internal consistency was assessed by calculating Cronbach’s alpha. 

The relationship between VL levels and sociodemographic characteristics was investigated through the distribution of the score by sex, age group, educational level, geographical area, level of financial deprivation, migration background, perceived social status, HL and health-related variables. An ordinal logistic regression (OLR), by which the sociodemographic factors were simultaneously analysed in association with the VL score, was also performed. The regression analysis was conducted using a proportional odds model, i.e., the effects of all explanatory variables are proportional to the various threshold values of the outcome variable. The suitability of a proportional odds logistic regression model depends on the assumption that each input variable has a similar effect on the various levels of the ordinal outcome variable. Such assumption was preliminarily tested using the Brant-Wald test.

Adopting the methodology proposed by VanderWeele and Ding [32], a sensitivity analysis was performed to evaluate how much residual confounding might be needed to explain away an effect estimate. For this purpose, the evalue Stata package proposed by Linden, Mathur and VanderWeele [33] was used.

Age was categorized in 4 age-groups: 18–28, 30–44, 45–64, and 65+ years of age.

The educational level was surveyed using the International Standard Classification of Education (ISCED) 2011 scale and classified into three levels: “low”, corresponding to lower secondary education or below (up to ISCED-2); “medium”, corresponding to higher secondary education (ISCED-3); and “high”, corresponding to post-secondary or short-cycle tertiary education (ISCED-4 and 5) or with a bachelor or higher level (ISCED-6 to 8).

Geographical areas were defined in accordance with the geographical subdivision adopted by the Italian National Institute for Statistics [34]: North-West (Liguria, Lombardy, Piedmont, Valle d’Aosta), North-East (Emilia-Romagna, Friuli-Venezia Giulia, Trentino-Alto Adige, Veneto), Center (Lazio, Marche, Toscana, Umbria), South (Abruzzo, Basilicata, Calabria, Campania, Molise, Puglia), and Islands (Sardinia, Sicily).

Three specific items were designed to investigate the respondents’ ability to afford medications, to afford medical expenses in general, and to make ends meet. The aim of these items was to derive a financial deprivation score calculated as the percentage (range 0–100) of items with valid responses that were answered with “very difficult” or “difficult”. The financial deprivation score was then categorised into the following levels: 0% = none, 0–33.33% = low, 33.34–66.66% = medium, and 66.67–100% = high.

Migration background was categorized in ‘No’ (no migration background), ‘one parent was born abroad’, ‘both parents were born abroad’, ‘born abroad’.

For the self-perceived social status variable, a scale in 10 categories (starting from the lowest perceived level ‘1’ to the highest perceived level ‘10’) was adopted.

The statistical analysis was performed using Stata 14 (StataCorp. 2015. Stata Statistical Software: Release 14. StataCorp, College Station, TX, USA).

## 3. Results

The internal consistency of the VL items was high (alpha = 0.81). For Italy, the mean VL score (i.e., the proportion of respondents answering “very easy” or “easy”) was 71.5%, while a mean of 28.5% of the Italian respondents indicated “very difficult” or “difficult” across all VL items. The items regarding judging vaccination information (COREHL26) and finding vaccination information (COREHL19) were rated the most difficult across countries [12]. In Italy, 33.3% of the 3327 valid answers to these two items indicated “very difficult” or “difficult”, while fewer Italians had difficulties in understanding why they or their family might need vaccinations (23.9%, COREHL22) and in deciding whether they should get a flu vaccination (23.4%, COREHL29) (Table 1).

As shown in Table 2, 67.6% of the respondents had a “good” (47.5%) or “sufficient” (20.1%) level of VL, while 32.4% had “limited” VL. In particular, the respondents with a prevalence of “limited” VL scores, which differed significantly from the average, were those from the south and islands (37.0%), those with a medium and high level of financial deprivation (37.9% and 48.7%, respectively), those with a medium and low educational level (32.2% and 33.4%, respectively), and those with some migration background (from 38.5% to 40.0%). An age over 65 years appeared to be associated with better performance, and there were no noteworthy sex differences in VL levels. 

The prevalence of the “good” level of VL increased with increasing age group, with increasing educational level, and with decreasing financial deprivation. The opposite happened with the prevalence of the “limited” VL level. When considering the prevalence of “good” and “sufficient” levels together, compared to “limited” levels, this relationship was maintained for education and financial deprivation, but was lost for age. No geographical gradient seemed to be evident for the “good” level of VL, although the highest prevalence of the “good” level was registered in the central geographical area, while the lowest prevalence was found in the South and Islands. This result remained true when contrasting the prevalence of “good” or “sufficient” levels with “limited” levels, given that 37.0% of the respondents in the South and Islands had “limited” levels of VL compared to a weighted mean of 30.1% for the other geographical areas.

In terms of the relationship between VL and HL, a positive association emerged between VL levels and HLS_19_-Q12 levels: the prevalence of “good” or “sufficient” VL scores increased in parallel with HLS_19_-Q12 levels, from 26.6%, in those with an inadequate HLS_19_-Q12 level, to 99.0% among those with an “excellent” HLS_19_-Q12 level. The opposite was true for the prevalence of “limited” VL scores, ranging from 73.5% in those with inadequate HLS_19_-Q12 levels to 1.0% among those with excellent levels. A similar association could be described between VL levels and HLS_19_-Q47 quartiles (Table 2): the prevalence of “good” level of VL increased with the increase in the HLS_19_-Q47 quartile; this trend remained considering “good” or “sufficient” VL levels together; the opposite occurred in the “limited” VL level.

In terms of the participants’ health status and use of health services, Table 3 suggests that VL is associated with self-assessed health status, the presence of long-term illness or health problems, limitations due to health problems and being trained in a healthcare profession. On the other hand, the VL results were not associated with the frequency of visits to a general practitioner or primary care physician. In particular, the prevalence of “good” or “sufficient” levels of VL decreased with reductions in self-assessed health status, ranging from 77.9% (63.8% + 14.1%) among those who rated their health as “very good” to 53.2% (34.4% + 18.8%) among those who rated it as “very bad”. Similarly, the prevalence of “good” or “sufficient” levels of VL was lower among those with at least one long-term illness or health problem (67.1% = 45.3% + 21.8%) compared with the healthy respondents (68.6% = 49.6% + 19.0%), particularly for the “good” level. The same was true among those in the “limited but not severely” (63.7% = 42.2% + 21.5%) or “severely limited” (56.9% = 41.5% + 15.4%) categories of the health status in terms of limitations due to health problems compared with the healthier respondents in the “not limited at all” category (71.3% = 51.1% + 20.2%); the opposite happens in the “limited” VL level. The percentage distribution of VL scores by VL sub-dimensions (corresponding to the four VL items) and by participant’ characteristics is shown in Appendix A.

Being trained in a healthcare profession appears to be associated with a better VL profile: among those who were trained, 53.7% have a “good” level of VL, compared to 46.6% among the lay individuals. This difference was slightly more marked for the VL “limited” level, for which trained respondents showed a prevalence of 24.5% compared to the 33.5% of their counterparts, while it was smaller for the VL “sufficient” level (21.8% vs. 19.9%).

Table 4 shows the results from the ordinal logistic regression analysis, including the odds ratios (OR) and 95% confidence intervals for the sociodemographic factors in relation to VL as the outcome variable. Given that the outcome variable is composed of three orderable modalities (“good” > “sufficient” > “limited”), according to the type of proportional odds model adopted, the resulting ORs can be interpreted as the odds of “limited” levels compared with the odds of “good” or “sufficient” levels for each unit increase/category change of each independent variable compared to its baseline.

Considering all social, economic and demographic factors in the same ordinal logistic model, only the financial deprivation categories and the “South and Islands” geographical area were significantly associated with VL levels, considering a 95% confidence interval. In particular, the odds of low VL levels (“sufficient” or “limited”) increased by 38% (13–68%), 86% (56–121%), and 209% (154–277%) in the respondents with a low, medium and high financial deprivation score, respectively, when compared to the individuals with no financial deprivation. Similarly, the odds of low VL levels were 25% (4–48%) higher for the individuals from the “South and Islands” geographical area compared to those from the “North-West” area.

### Sensitivity Analysis

The observed OR of 3.09 for high deprivation could be explained away by an unmeasured confounder that was associated with both the explaining factors and the outcome by a risk ratio of 5.53-fold each, above and beyond the measured confounders, but weaker confounding could not do so [32]. Similarly, the E-value for the lower confidence limit (LCL) is 4.36, which can be interpreted as “unmeasured confounders” associated with outcome and explaining factors by an OR of 4.36-fold each could explain away the lower confidence limit, but weaker confounding could not. As well as for the OR of 1.86 for “medium” level of deprivation, with an E-value point estimate of 3.04 and an E-value for the LCL of 2.09, the evidence for causality from these E-values thus looks reasonably strong because substantial unmeasured confounding would be needed to reduce the observed association or its Confidence Interval to null. On the other hand, the sensitivity analysis for the OR associated with a “low level” of financial deprivation and the residence in the South and Islands regions did not provide robust evidence of statistical significance, as the E-values for the LCL were too low and little unmeasured confounding factors would be sufficient to nullify the estimated effect. 

## 4. Discussion

In this work, a specific tool for measuring VL (as, for instance, that of Biasio et al. [20] or the tool proposed in the same M-POHL survey that was adopted as an additional package by certain countries) was not used. The analysis performed here focused on the four items related to vaccination and included in the HLS_19_ questionnaire, each referring to one of the key topics of HL (access, understand, appraise, and apply). This is undoubtedly a major limitation of our study, but it also represents one of its main strengths. In fact, the simplicity of our design allowed us to estimate VL in a large representative sample of the Italian population, which would have been hard, if not impossible, to reach using specific tools for VL. Moreover, the use of four items integrated in a broader survey allowed us to evaluate the major determinants of VL, while also adjusting for potential confounders. 

Overall, certain important considerations for Italian respondents can be derived from the results. First, the Italian adult population appears to have more confidence in their own and their family’s decisions to get vaccinated and act accordingly, rather than finding information and processing or judging information and vaccination needs. This approach seems to reflect a trend towards an empowerment not completely supported by literacy (and knowledge) itself, which could represent a certain danger in the vaccination decision process. An example can be represented by the phenomenon of anti-COVID vax and vaccine-hesitant individuals, who comprise approximately 10% of the Italian population.

Moreover, this study confirms numerous international studies that demonstrated an association between health literacy (in the domain of vaccination in this specific case) and social, economic, and demographic determinants. In this study, those determinants included financial deprivation and living in the southern part of the country or in its major islands. These were the two characteristics significantly associated with the lowest VL levels, even though only the ‘high’ and ‘medium’ levels of financial deprivation were supported by the sensitivity analysis, confirming the results of international and national studies on HL and, more generally, on health issues [35,36,37]. Indeed, in all countries included in the recent HLS_19_ survey, except for Bulgaria, financial deprivation was the strongest predictor of VL in a multivariable linear regression model: those with high financial deprivation had a lower VL level compared with the not deprived subpopulation [12].

According to this study results, the VL level was not significantly associated with gender, age and educational level in the Italian respondents, a result partially confirmed by the multivariable linear regression model presented in the HLS_19_ international report, which found no statistically significant association with gender and education and only a small statistically significant association with age (rho = 0.05, *p* < 0.05). More generally, a statistically significant association between VL and age was reported for only 4 of the 11 countries that administered the 4 VL items in HLS_19_ (Ireland, Italy, Norway and Portugal), while a more widespread effect was observed in terms of education (Bulgaria, Czech Republic, Germany, Ireland, Norway, and Slovenia) [12]. Despite not achieving statistical significance in the logistic regression model, it is worth noting that the estimated ORs for a poorer VL level showed a decreasing trend with increasing age and higher education, a result that confirms the insights from our descriptive analysis and from the HLS_19_ international report on the role of age and education in VL. The statistical significance of their effect, however, might be lost due to collinearity with financial deprivation.

The current study reveals that VL levels were lower in individuals with lower socioeconomic status, such as those with average to very poor self-perceived health status and low self-perceived social status. In several countries (Austria, Bulgaria, Hungary, Ireland, Italy, Norway, Portugal and Slovenia), individuals with a “poor” self-perceived health status reported lower VL levels compared with those with a very good health status [12]. Interestingly, higher rates of self-perceived social status in this study seemed to be proportionally associated with higher VL scores only if we do not consider the two lower and the two higher categories, a result that is possibly due to the low sample number for these categories and to the so-called “extreme aversion bias” by which respondents might avoid choosing the extremes of a scale in their answers. 

Furthermore, having a migration background did not seem to be systematically associated with lower VL levels. Having only one parent born abroad was linked to a lower prevalence of a “good” level (36.9%) of VL score when compared with having both parents born abroad (50.0%) or being born abroad (40.4%). Unexpectedly, having both parents born abroad was associated with a higher prevalence of a “good” VL score even when compared with having no migration background (47.8%). However, this effect was partially lost when considering the “good” and “sufficient” VL aggregated levels in comparison with the “limited” level. In this case, even though based on a limited number of cases, having both parents born abroad appeared to be associated with a poorer VL outcome compared with those with only one parent born abroad or being born abroad. Overall, this result is encouraging in terms of an open society, in which citizenship (and access to services and the right to health) should not be linked to the place of origin of the individual or of their parents. However, an analysis differentiated by country of origin might show a different role of migration background depending on the respondents’ specific provenance. Furthermore, as mentioned in the HLS_19_ international report, immigrants with poor Italian language skills might not have been included in the sample [12]. These possibilities suggest the need for further studies focusing on migrants, with larger sample sizes and administering the questionnaire in the respondents’ native language.

A clear association was shown between VL and HL, measured both as quartiles of the results from the HLS19-Q47 questionnaire and from its 12-item short form, outcome that reveals in both cases a two-sided effect: the higher the HL level, the higher the percentage of “good” or “sufficient” VL; the lower the HL level, the higher the percentage of “limited” VL. This result was expected, but its confirmation is important in terms of designing and implementing interventions aimed at increasing HL. Such interventions could also affect decisions regarding getting vaccinated against highly contagious and virulent biological agents, as in the case for flu and COVID-19.

Regarding specific training for healthcare professionals, the data are quite disappointing. Although there was a slightly higher percentage of respondents with a “good” or “sufficient” VL among those who had been trained in the healthcare sector, 24.5% of trained staff had a “limited” level of VL. This finding suggests more attention is needed to the curricula of health professionals on HL and VL, particularly because these workers represent one of the main trusted sources of information for the general population in terms of health-related advice [38]. Healthcare professionals can be considered role models for promoting healthy lifestyle behaviors and protective behaviors during pandemics, given that vaccine hesitancy can represent a major barrier to curbing the spread of COVID-19 [39].

Another aspect highlighted by the current study (and confirmed in the HLS_19_ international report) is that VL items related to finding and judging vaccination information are the most difficult for the respondents [12], which is related not only to the individual’s ability to find and understand the information but also to the health communication skills of the actors involved in conveying scientific information to the general population. This difficulty also leads to misinformation/disinformation in coronavirus-related information, doubts regarding policy recommendations, and vaccine hesitancy. There is, therefore, an urgent need for targeted strategies to improve the communication skills of members of the scientific community and the media [40,41,42]. 

The study presents certain limitations: first, it is a cross-sectional study conducted over 1 month and by means of online and telephone interviews. The number of replacements (i.e., the individuals who refused to be interviewed before reaching the expected number of 3500 participants) is unknown. Moreover, it is difficult to estimate the confirmation bias linked to the interviewees’ willingness to declare abilities and competencies they do not possess, which is typical of all subjective tools. As already mentioned, we did not use a dedicated VL tool and limited the analysis to the extrapolation of the four vaccination-related items from the general HLS_19_-Q47 questionnaire. From this point of view, further research is needed to evaluate to what extent the four item scale we used is able to predict VL measurements made with other validated tools.

The study also presents several strengths: it is the first time Italy has participated in a general and standardised European survey assessing HL and VL. The tool employed was the same for all 17 countries that adhered to the M-POHL international network and implemented the HLS_19_ survey, thereby allowing comparisons within the European Union. The Italian sample was representative of the general adult population aged 18 years and over and included men and women from all geographical areas of the country, as well as from all age groups (with a slight preponderance of 45–64 year-olds). Lastly, the results are promising in terms of offering elements to policymakers to help establish interventions on public HL as a civil right and to help make evidence-based decisions for improving the population’s health and quality of life.

## 5. Conclusions

The main results of this research highlight the complexity of dealing with vaccination information in Italy, both in absolute terms and in relation to other countries. Across all countries, demographic and socioeconomic factors were found to be determinants of VL, with lower VL scores associated with lower educational level, lower perceived social status, higher financial deprivation and lower self-assessed health status [12]. The research applied to the Italian context confirmed an association between VL and financial deprivation while also revealing a disadvantaged VL level for southern geographical areas. Although not statistically significant in the regression model, age and education also played a plausible role in determining vaccination-related health literacy, as well as an important association between VL and other factors, such as migration background and health-related variables. These associations deserve further exploration in Italy. To conclude, although further analysis is needed, improving VL in Italy and in other countries should be a top priority in the political agenda, with a special focus on the social gradient involved. In fact, population health policies need to move toward a proportionate universalism to drive the most appropriate solutions within the social gradient and to increase efficiency while ensuring equity. Further research is needed on implementing evidence-based plans to support national and local actions in favor of developing multitargeted information literacy. This study contributes to highlight the possible solutions based on population vaccination-related health literacy.

## Figures and Tables

**Table 1 ijerph-19-04429-t001:** Percentage distribution of the valid answers by each single item and subsection of the VL scale.

Code	Sub-Dimension	On a Scale from “Very Easy” to “Very Difficult”, How Easy or Difficult Would You Say It Is:	Very Difficult	Difficult	Easy	Very Easy
COREHL19	Find	to find information on recommended vaccinations for you or your family?	4.5	28.8	53.0	13.7
COREHL22	Understand	to understand why you or your family may need vaccinations?	3.5	20.4	53.8	22.3
COREHL26	Judge	to judge which vaccinations you or your family may need?	4.9	28.4	52.1	14.6
COREHL29	Decide	to decide if you should have a flu vaccination?	3.5	19.9	54.5	22.1

**Table 2 ijerph-19-04429-t002:** VL levels by respondents’ variables related to socioeconomic background and HL. *n* = number of respondents for each category; % = percentage of respondents for each category.

Variable	Category	*n*	%	Vaccine Literacy Levels (%)		
				Good	Sufficient	Limited
Validity of response	Valid	3500	100.0	47.5	20.1	32.4
Sex	Male	1685	48.1	47.0	20.7	32.3
	Female	1815	51.9	47.9	19.7	32.4
Age	18–29	468	13.4	45.2	22.3	32.5
	30–44	826	23.6	45.7	19.1	35.2
	45–64	1254	35.8	47.5	19.1	33.4
	65+	952	27.2	50.0	21.4	28.6
Education	Low	1470	42.0	46.0	20.6	33.4
	Medium	1299	37.1	46.3	21.5	32.2
	High	731	20.9	52.5	16.8	30.7
Geographical area	North-West	939	26.8	50.2	19.3	30.5
	North-East	710	20.3	47.8	22.2	30.0
	Center	681	19.5	51.2	19.3	29.5
	South and islands	1170	33.4	42.9	20.1	37.0
Financial deprivation (missing values = 188)	None	1478	44.6	57.8	18.3	23.9
	Low	539	16.3	47.7	22.3	30.0
	Medium	743	22.4	41.6	20.5	37.9
	High	552	16.7	27.9	23.5	48.6
Migration background	No	3353	95.8	47.9	20.0	32.1
	One parent was born abroad	72	2.0	36.9	24.6	38.5
	Both parents were born abroad	10	0.3	50.0	10.0	40.0
	Born abroad	65	1.9	40.4	22.8	36.8
Self-perceived social status (missing values = 54)	1 (lowest)	54	1.6	34.7	14.3	51.0
	2	67	1.9	49.2	25.4	25.4
	3	176	5.1	32.5	23.7	43.8
	4	308	8.9	42.4	18.2	39.4
	5	656	19.1	42.8	18.9	38.3
	6	1081	31.4	46.8	21.9	31.3
	7	783	22.7	54.8	20.1	25.1
	8	253	7.3	58.5	21.6	19.9
	9	43	1.3	53.7	2.4	43.9
	10 (highest)	25	0.7	24.0	12.0	64.0
HLS_19_-Q47 quartile	First	896	25.6	6.4	15.1	78.5
	Second	854	24.4	32.2	30.7	37.1
	Third	875	25.0	57.4	30.3	12.3
	Fourth	875	25.0	94.1	5.3	0.6
HLS-Q12 level (missing values = 29)	Inadequate	802	23.1	9.3	17.3	73.4
	Problematic	1201	34.6	35.8	28.7	35.5
	Sufficient	1157	33.3	73.1	18.2	8.7
	Excellent	311	9.0	95.7	3.3	1.0

**Table 3 ijerph-19-04429-t003:** VL levels by respondents’ variables related to health status and healthcare services use. *n* = number of respondents for each category; % = percentage of respondents for each category.

Variable	Category	*n*	%	Vaccine Literacy Levels (%)		
				Good	Sufficient	Limited
Validity of response	Valid	3500	100.00	47.5	20.1	32.4
Self-perceived health status (missing values = 12)	Very good	220	6.3	63.8	14.1	22.1
	Good	1423	40.8	53.0	20.1	26.9
	Fair	1592	45.6	42.0	21.4	36.6
	Bad	218	6.3	39.5	17.6	42.9
	Very bad	35	1.0	34.4	18.8	46.8
Long-term illness or health problems (missing values = 85)	None	1910	55.9	49.6	19.0	31.4
	One or more long-term	1505	44.1	45.3	21.8	32.9
Limitations due to health problems (missing values = 171)	Severely limited	285	8.6	41.5	15.4	43.1
	Limited but not severely	1099	33.0	42.2	21.5	36.3
	Not limited at all	1945	58.4	51.1	20.2	28.7
Training in a healthcare profession	No	3070	87.7	46.6	19.9	33.5
	Yes	430	12.3	53.7	21.8	24.5
Visits to GP in 12 months (missing values = 335)	0–4	2562	80.9	49.7	19.6	30.7
	5–9	383	12.1	52.4	20.5	27.1
	10–14	164	5.2	68.3	9	22.7
	15–19	9	0.3	8.3	8.3	83.4
	20–24	31	1.0	73.2	10.3	16.5
	25–29	6	0.2	20	20	60
	30–34	9	0.3	25	0	75
	35–39	1	0.0	0	100	0

**Table 4 ijerph-19-04429-t004:** Association between VL main score level and participant characteristics (sex, age, educational level, financial deprivation, perceived social status, geographical area) in terms of odds ratio (OR) and related 95% confidence interval (95% CI) limits with sensitivity tests for statistically significant ORs (E-value). PE = point estimate; LCL = lower confidence limit.

Variable	Category	OR	95% CI Lower Limit	95% CI Upper Limit	E-Value (PE)	E-Value (LCL)
Sex	Male (reference)	1.00				
	Female	0.90	0.79	1.03		
Age	18–29 (reference)	1.00				
	30–44	1.03	0.80	1.30		
	45–64	0.92	0.74	1.15		
	65+	0.82	0.64	1.04		
Education	Low (reference)	1.00				
	Medium	0.96	0.82	1.27		
	High	0.89	0.73	1.08		
Financial deprivation	No (reference)	1.00				
	Low	1.38	1.13	1.68	2.15	1.56
	Medium	1.86	1.56	2.21	3.04	2.09
	High	3.09	2.54	3.77	5.53	4.36
Geographical area	North-West (reference)	1.00				
	North-East	1.07	0.87	1.29		
	Center	0.97	0.79	1.18		
	South and Islands	1.25	1.04	1.48	1.81	1.24
Threshold of the outcome	/cut1	0.19	−0.08	0.45		
	/cut2	1.07	0.79	1.35		

## Data Availability

The dataset generated and analysed during the current study is available from the corresponding author on reasonable request, according to the Data Protection Officer of the Istituto Superiore di Sanità, Rome, Italy.

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
