# Peer review of "The Determinants of Vaccine Literacy in the Italian Population: Results from the Health Literacy Survey 2019"

_ijerph, 2022, doi:10.3390/ijerph19084429_

Round 1

Reviewer 1 Report

The paper addresses an interesting and relevant topic such as the evaluation of the determinants of vaccine health literacy in Italy. The manuscript is well written, quite clear and complete.

Tables are not easy to follow 

Author Response

Dear reviewer,

thank you for your comments and assessment of our work.

We modified the tables making them more readable. We also let the whole article be reviewed in English for minor spell check. 

Reviewer 2 Report

Thank you for the opportunity to revise this manuscript. The manuscript investigates the determinants of vaccination health literacy in a sample of Italian adults.

The paper has several flaws and I do not consider it suitable for publication on IJERPH in its current form.

Here some major concerns:

The topic is vaccination health literacy. I am not sure where this comes from. Is it vaccine literacy? Is it health literacy? Reference number 1 and 2 that are used to define vaccination health literacy regard health literacy in general terms, so they are not appropriate.

Vaccine literacy is usually measured using specific scales, that need to be tested and validated. In this paper, the authors selected a posteriori three questions from a previous survey and created a “vaccination health literacy” scale, with no validation and comparison with other existing instruments about the reliability of the tool. In addition, how the authors used participants’ answers to categorize vaccination health literacy into three classes needs to be further explored. The methods used are not sufficient, a sensitivity analysis should be performed and it is not clear how the variables were selected to build a model. Therefore, every analysis that follows these considerations is meaningless.

I would be careful about defining health literacy as excellent. Probably high or good is enough.

References are outdated. There are recent systematic reviews on health literacy that should be mentioned.

Author Response

Dear reviewer, 

thank you for your valuable reading of our manuscript. 

We have taken into account your suggestions and amended the article. 

In the Word attached you can find the point-by-point answers to your comments. 

Reviewer 3 Report

The current study is on a topic of relevance and importance for this journal. While overall, I found the paper to be well written, there are some examples of poor phrasing throughout which will need clarification. I feel confident that that authors have used an appropriate study design and performed appropriate statistical analyses of the data set.   

The paper can be improved by providing more clarity in the Introduction on the term HL-VAC. . Its first mention needs to be referenced.  In the introduction it is not clear if this is a new term (is so a ref is needed) or one coined by the authors of this paper. In fact some information that is provided later, in the discussion, does address these issues. Therefore, information in the discussion section re definitions of HL-VAC can be moved to the introduction.

Line 74: Provide a reference for ‘a recent systematic review’. Ref 13 in the ref list has not year included.

Materials and methods:

Line 113. Ref to Table 1 here to view these 4 items.

Results:  The sub dimensions listed in table 1 have not been explained in the paper.

Discussion-as noted earlier the literature on defining HL-VAC should be in the Introduction.

Line 289: Phrasing: change to our study confirms much of the research which supports

Ref number 24 does not appear to be correctly cited.

Line 374-meaning of who do not possess is not clear. It appears that there is a word missing here.

Line 380 lower case for countries.

Line 382: change end to and.

Author Response

Dear Reviewer,

thank you for all your valuable comments on our manuscript.

Please find written in red in the attachment the point-by-point answers to your requests.

Round 2

Reviewer 2 Report

The manuscript has improved notably, however the bibliography is still poor and should be updated. There are several self-citations in the manuscript, but please check carefully any other relevant paper.

Author Response

Dear reviewer,

thank you for your comment.

We have updated the bibliography with more recent references, especially with regard to international studies. We were not able to significatively reduce the presence of self-citations, due to the pivotal role of some of them in this research field and/or to lack of other studies concerning vaccine literacy in the Italian setting. 
